# Soft Tissue Sarcoma Study: Association of Genetic Alterations in the Apoptosis Pathways with Chemoresistance to Doxorubicin

**DOI:** 10.3390/cancers14071796

**Published:** 2022-04-01

**Authors:** Evgeny M. Kirilin, Timur I. Fetisov, Natalia I. Moiseeva, Ekaterina A. Lesovaya, Lidia A. Laletina, Leyla F. Makhmudova, Angelika E. Manikaylo, Liliya Y. Fomina, Denis A. Burov, Beniamin Yu. Bokhyan, Victoria Y. Zinovieva, Alice S. Vilkova, Larisa V. Mekheda, Nikolay A. Kozlov, Alexander M. Scherbakov, Gennady A. Belitsky, Vytas Švedas, Kirill I. Kirsanov, Marianna G. Yakubovskaya

**Affiliations:** 1Belozersky Institute of Physicochemical Biology, Lomonosov Moscow State University, Lenin Hills 1, 119991 Moscow, Russia; kirilin@belozersky.msu.ru; 2N.N. Blokhin NMRCO, 115478 Moscow, Russia; timkatryam@yandex.ru (T.I.F.); n.i.moiseeva@gmail.com (N.I.M.); lesovenok@yandex.ru (E.A.L.); panlidia@gmail.com (L.A.L.); mahmusha@yandex.ru (L.F.M.); 7717271@mail.ru (A.E.M.); 3050244@gmail.com (L.Y.F.); denisburov@yandex.ru (D.A.B.); beniamin-bokhyan@mail.ru (B.Y.B.); vichka2396@gmail.com (V.Y.Z.); cunning.fox@mail.ru (A.S.V.); lmeheda@gmail.com (L.V.M.); newbox13@mail.ru (N.A.K.); alex.scherbakov@gmail.com (A.M.S.); belitsga@mail.ru (G.A.B.); kkirsanov85@yandex.ru (K.I.K.); 3Department of Oncology, I.P. Pavlov Ryazan State Medical University, 390026 Ryazan, Russia; 4Research Computing Center, Faculty of Bioengineering and Bioinformatics, Lomonosov Moscow State University, Lenin Hills 1, 119991 Moscow, Russia; vytas@belozersky.msu.ru; 5Institute of Medicine, RUDN University, 117198 Moscow, Russia

**Keywords:** soft tissue sarcoma, undifferentiated pleomorphic sarcoma, synovial sarcoma, chemoresistance, Doxorubicin, Ifosfamide, Gemcitabine, Docetaxel, genetic alterations, apoptotic signaling

## Abstract

**Simple Summary:**

Genotoxic chemotherapy is the main component of the treatment for advanced soft tissue sarcomas. However, its efficacy is rather low and it is followed by rapid appearance of drug resistance. Our study was directed to the search of molecular drivers of chemoresistance in synovial and undifferentiated pleomorphic sarcomas to genotoxic drugs mostly used for their treatment. Using primary cell cultures obtained from sarcomas after surgery, we estimated their chemoresistance in vitro and performed exome sequencing. We revealed that cancer cells of more than one quarter of patients had molecular alterations preventing apoptosis and observed an association between molecular alterations found and chemoresistance to Doxorubicin, but not to Ifosfamide or Gemcitabine and Docetaxel. Information concerning the peculiar drivers of individual drug resistance could help to improve personalized chemotherapy by withdrawal from an inefficient drug or by targeting the revealed mechanism of chemoresistance.

**Abstract:**

Soft tissue sarcomas (STS) are heterogeneous cancers with more than 100 histological subtypes, different in molecular alterations, which make its personalized therapy very complex. Gold standard of chemotherapy for advanced STS includes combinations of Doxorubicin and Ifosfamide or Gemcitabine and Docetaxel. Chemotherapy is efficient for less than 50% of patients and it is followed by a fast development of drug resistance. Our study was directed to the search of genetic alterations in cancer cells associated with chemoresistance of undifferentiated pleomorphic and synovial sarcomas to the abovementioned genotoxic drugs. We analyzed chemoresistance of cancer cells in vitro using primary STS cultures and performed genetic analysis for the components of apoptotic signaling. In 27% of tumors, we revealed alterations in *TP53, ATM, PIK3CB, PIK3R1, NTRK1*, and *CSF2RB*. Cells from STS specimens with found genetic alterations were resistant to Dox, excluding the only one case when TP53 mutation resulted in the substitution Leu344Arg associated with partial oligomerization loss and did not cause total loss of TP53 function. Significant association between alterations in the components of apoptosis signaling and chemoresistance to Dox was found. Our data are important to elaborate further the therapeutic strategy for STS patients with alterations in apoptotic signaling.

## 1. Introduction

Soft tissue sarcomas (STSs) are the heterogeneous group of malignant tumors of mesenchymal origin that account for 1% of adult solid tumors [1]. STSs are characterized by aggressive disease progression in the absence of relevant treatment. According to clinical guidelines, adjuvant/neoadjuvant chemotherapy based on anthracyclines may be a reasonable option for some groups of STS patients with stage 2–3 and it is recommended as the first-line treatment for advanced/metastatic, clinically unresectable STS [2]. However, 2-year overall survival of patients with advanced/metastatic STS treated with doxorubicin and Ifosfamide is 31%, while complete response to chemotherapy is observed for 2% of patients, partial response for 25%, and stabilizing effect for 50% [3]. It is explained by the limitation in understanding of the molecular genetics and biology of these cancers. According to WHO classification, STSs include more than 100 nosological entities with different histogenesis, which are expected to possess molecular peculiarities in genome function [4]. Methodological progress in genetic investigations opens up opportunities for identification of molecular alterations with prognostic and predictive significance for clinic. Our study was directed to the search for molecular determinants of chemoresistance in synovial sarcoma (SS) and undifferentiated pleomorphic sarcoma (UPS). We analyzed effects of Doxorubicin (Dox), Ifosfamide (Ifo), Gemcitabine (Gem), and Docetaxel (Doc), which are included in the first and second chemotherapy lines of STS [2].

Dox activity is determined by the inhibition of DNA replication and transcription caused by the drug intercalation into DNA, inhibition of topoisomerase II, and formation of highly reactive free radicals, which induce DNA damage. DNA damage, in turn, stimulates the cell cycle arrest and apoptosis activation [5,6]. Ifo belongs to the class of the alkylating agents [7], and Gem is the antimetabolite from the pyrimidine antagonists [8]. Doc inhibits microtubule depolymerization, resulting in cell cycle arrest in G2 and M phases, as well as in aneuploidy [9]. Thus, the most effective chemotherapeutics for STS treatment are genotoxic agents.

TP53 plays a key role in cell response to DNA damage, cell cycle regulation, DNA repair, and induction of cell death [10]. TP53 mutations are the most frequent genetic alterations in STSs [1,11,12]. It was demonstrated that the leiomyosarcoma cells SK-LMS-1 with mutated TP53 are less sensitive to Dox than non-mutated cells [13]. These data were confirmed by Lee et al. on a number of cell lines, including osteosarcoma cells MG63 and U2OS. It was shown that Dox resistance of the studied cells is associated with TP53 mutations [14].

TP53-mediated chemoresistance is explained by the disruption of apoptotic signaling. Evidently, chemoresistance determined by inactivation of apoptosis signaling should be caused not only by TP53 abnormalities, but also by mutations in the other genes. Moreover, the role of structural alterations in the main gene of the signaling pathway is complicated to reveal in studies of cancers with a low level of mutations in small cohorts of patients. Complex analysis of many components of the apoptotic signaling simultaneously could be a better option to analyze chemoresistance determinants.

Therefore, the primary aim of our study was the analysis of gene profiling associated with apoptosis pathways in STS specimens from patients with SS and UPS by exome sequencing. Simultaneously, we analyzed drug resistance of STS cells by in vitro chemosensitivity/chemoresistance assay (CSRA). We used CSRA proposed by Kurbacher et al. [15], which, at present, is used in many modifications [16,17,18,19], based on different methods for estimation of cell viability [20,21,22]. We then analyzed the association between chemoresistance and alterations in the genes of apoptosis pathways.

## 2. Materials and Methods

### 2.1. Tumor Specimens

STS specimens were obtained from patients who had soft tissue sarcoma and underwent surgery at the N.N. Blokhin National Medical Research Center of Oncology between 2018 and 2020. The average age of patients was 47.8 ± 15.7 years, and there were 19 males and 18 females. The STS stages were distributed as follows: I-1 (3%), II-7 (19.0%), III-27 (72%), and IV-2 (6%). We analyzed 25 UPS and 12 SS.

### 2.2. Primary Cancer Cells Cultures and Chemosensitivity Assay

CSRA was performed as a routine procedure immediately following surgery. Solid tumors were obtained during surgery and cut into smaller fragments (1 mm^3^), which were then dissociated to prepare suspensions of single cells by the incubation in 5–10 mL sterile collagenase mixture for 2–3 h at 37 °C on a shaker. One part of the cells was used for CSRA, and the other part was used for sequencing, followed by the exome analysis. The cell suspension (1–2 × 10^5^ cells/mL, 100 μL) was added to each well of a 96 well microplate. The efficiency of establishing primary cell cultures from tumor tissue was 81%. We analyzed chemoresistance of cultured STS cells to Dox (RONC, Russia), Doc (NATIVA, Russia), Gem (BIOCAD, Russia), and Ifo metabolite 4-hydroperoxy-Ifosfamide (NIOMECH, Germany), which was used as Ifo, a prodrug that requires in vivo hepatic activation. Tested chemotherapeutic drugs were added at six different concentrations: 6.25, 12.5, 25, 50, 100, and 200% of a standard test drug concentration (TDC) [23]. The TDCs were chosen according to pharmacokinetic data for standard doses of the agents, adjusted to give good discrimination [21]. In particular, we used drug concentrations presented in the Table 1.

Triplicate wells with untreated cells served as a vehicle control. The wells containing only culture medium provided the blank for the absorbance readings. Highly sensitive STS primary cultures demonstrating a dose-response were used as a positive control in every CSRA performance (AFN45 for Dox, Ifo, Dox + ifo, Doc, Doc + Gem, and AFN41 for Ifo, Dox + ifo, Doc, Gem, and Doc + Gem). Plates were incubated for 5–6 days at 37 °C, with 95% humidity in a 5% CO_2_ incubator. Cell viability was measured using resazurin-based assay, as described previously [22]. The results of CSRA were interpreted using the sensitivity index SI (SI = 600—sum of % inhibition at 200, 100, 50, 25, 12.5, and 6.25% TDC) [23].

### 2.3. Cytology for STS Primary Cultures

We determined the percent of STS cells in all primary cultures used in CSRA. Thin-layer slide preparation was performed using the Thermo Shandon Cytospin 3 (Marshall Scientific, Hampton, NH, USA). Morphological assay of Leishman-stained slides was per-formed on a microscope «Nikon Eclipse Ci-S» (Nikon Corporation, Tokyo, Japan) at 500-x magnification in three fields of view. The percentage of malignant cells was calculated using traditional cytological criteria, including cell shape, architecture, and characteristics related to genomic instability [24].

### 2.4. Bioinformatic and Statistical Analysis

Exomes for the 37 patients were captured with Agilent SureSelect Focused Exome with an average coverage of at least 100× for each sample and a pair-end read length of at least 250 nucleotides. All reads were trimmed for adapter sequences and low-quality reads with a large number of unknown nucleotides using fastp program [25]. Sequence reads were aligned to the GRCh38 human reference genome using the Burrows-Wheeler method implemented in BWA mem [26] and followed by marking duplicates and recalibrating base qualities by GATK4 methods [27]. Tumor-only variant calling was performed on tumor samples with no paired normal tissue using in-house panel of normals and utilizing advantages of normal cell contamination implemented in GATK4 Mutect2 probabilistic models for genotyping and filtering. Primary annotation of vcf files was performed using SnpEff and SnpSift (v4.3) [28] with standard parameters, as well as GATK4 Funcotator. Additional annotation was carried out by the AnnotSV program [29], which combines comprehensive additional information (OMIM, Gene intolerance, Haploinsufficiency, DGV, 1000genomes, pathogenic structural variations from dbVar, etc.). Significantly mutated genes were identified using MuSiC program [30]. We analyzed the genes of apoptosis signaling proteins described in the KEGG PATHWAY database, available at https://www.kegg.jp/pathway/hsa04210 (last accessed date: 5 February 2022). R version 4.1.2 was used for statistical analysis, including one-sided Fisher’s exact tests, along with R/Bioconductor package maftools [31] to visualize and analyze mutational data.

## 3. Results

We analysed 37 primary cultures obtained from STS specimens from 25 patients with UPS and 12 patients with SS. Clinical parameters of patients are summarized in Table 2.

We evaluated the sensitivity index (SI) of STS cells to Dox, Ifo, Doc, Gem, and their combinations (Dox + Ifo и Doc + Gem) using CSRA, as described earlier [15,30]. According to this approach, lower SI corresponds to the higher sensitivity of cells. At SI values lower than 250, we defined cells as sensitive to the drug or drug combinations. Application of Dox and Ifo corresponds to tumor cell resistance (SI > 250) of 71% and 64%, respectively, while 43% of tumor specimens were resistant to the Dox + Ifo combination. In total, 68% and 61% of tumor samples were resistant to Doc and Gem, respectively, and 45% of STS samples were resistant to their combination. Figure 1 presents the histograms of SI frequencies for all the drugs studied individually and in combination. Primary STS cell cultures demonstrated the high heterogeneity in the response to drug exposure, while SI frequencies did not correspond to the normal distribution.

Studied cohort of STS specimens is characterized by a low level of mutations (1.02 for Mb). MuSiC program identified TP53 as the only significantly mutated gene (compared to the expected mutation level for the entire cohort, FDR < 0.05). Aiming for elucidation of a common resistance mechanism within an entire cohort with a substantial number of resistive samples, we focused on analysis of TP53 structural variations, as well as other proteins participating in the apoptosis pathway KEGG_APOPTOSIS (Figure 2).

Revealed *TP53* mutations represent nonsense mutations in the specimens AF50b, AFN119b, and AFN120b, frameshift in specimen AF93b, and point substitutions in AF53b and AF98b (Table 3).

All abnormalities are characterized in the COSMIC database as pathogenic. *ATM* gene mutation revealed in specimen AFN111b is also defined as pathogenic. Abnormalities in *PIK3R1* and *PIK3CB* genes were demonstrated in the specimens AFN128b and AFN80b, respectively. Although COSMIC does not provide information concerning the exact substitutions of amino acids found in our study, it defines the alternative substitutions in the corresponding positions as pathogenic.

Tumor cells from STS specimens with the alterations in *TP53, ATM, PIK3CB, PIK3R1, NTRK1*, and *CSF2RB* genes were resistant to Dox (association *p*-value = 0.03, Fisher’s exact test), excluding specimen AF98b. Association of tumor cell resistance to Dox with genetic alterations in apoptosis signaling was not demonstrated for UPS or SS. Interestingly, among the seven observed genetic abnormalities in UPS, six mutations were in TP53 and one in CSF2RB. This means that TP53 mutations were frequent in UPS. We did not find any TP53 abnormalities in SS, and all the found mutations were in the other genes of the apoptosis signaling (3 out of 3), (*p* = 0.03). We did not reveal the association of tumor cell resistance to Doc and Gem with abnormalities in genes from the apoptosis signaling (Table 4, Figure 3).

## 4. Discussion

The complexity of sarcoma research is associated with a high diversity of histological characteristics, as well as genetic peculiarities underlying their development. In one of the last big studies, the molecular landscape of 206 STS samples from 6 major sarcoma subtypes was obtained using multiplatform analysis [1]. We studied less characterized STS subtypes—UPS and SS. The analyzed cohort included 12 SS specimens and 25 UPS specimens. When analyzing mutations in genes associated with apoptotic signaling, the maximum number of mutations was found in the *TP53* gene.

*TP53* mutations could lead to either total/partial loss-of-function of tumor suppressor protein or gain-of-function as TP53 is multifunctional and it is involved in the regulation of different signaling pathways [32]. Most mutations in studied STS specimens were *TP53* nonsense mutations inducing early translation breakage and synthesis of truncated protein. In specimens AF50b and AFN120b, translation breakage was in codons, corresponding to amino acids Tyr126 and Lys132, leading to an incomplete synthesis of a DNA-binding domain and loss of oligomerization domain. In specimen AFN119b, translation breakage near Arg342 also leads to protein loss-of-function, as demonstrated in vitro by a decrease of TP53 nuclear localization, as well as an absence of transactivation [33]. A frameshift after Glu258 residue in DNA-binding domain in specimen AF93b also leads to loss-of-function because of incorrect synthesis of the functional region of this domain and loss of the oligomerization domain. H193Y mutation in the DNA-binding domain of *TP53,* revealed in specimen AF53b, according to Clarke et al., is associated with lower mRNA expression of CDKN1A and protein expression of Fdxr [34]. Moreover, Oliveira et al. refer this mutation to abnormalities associated with gain-of-function and leading to TP53 accumulation [35]. Accumulation of the mutated TP53 in the case of gain-of-function mutations do not ensure the high transcriptional activity similar to wild type protein but lead to the activation of different signaling pathways [32]. Leu344Arg mutation found in specimen AF98b is located in the tetramerization domain and leads to a decrease of tetramerization ability and protein loss-of-function. Mutation in Leu344 position could define a different level of oligomerization: Leu344Pro mutation is associated with total loss of oligomerization, while Leu344Arg mutation is associated with partial oligomerization loss [36]. Partial maintenance of TP53 function could be the reason for sensitivity to Dox in specimen AF98b.

We found no *TP53* mutations in specimen AFN111b; however, we revealed the substitution Arg2691Cys in ATM that makes it unable to phosphorylate TP53 at Ser15. This TP53 post-translational modification appears in the response to DNA damage [37]. Structural modeling demonstrated that Arg2691Cys mutation affects the physicochemical properties of the ATP-binding pocket. Disruptions in the structure of the element critical for kinase function lead to TP53-related loss-of-function [38].

Thus, all known mutations of the components from TP53-mediated apoptotic signaling found in studied specimens lead to disruption of its function, excluding the sample AF98b. Leu344Arg mutation in TP53 results in the decrease but not loss of functional activity. Analysis of the association between chemoresistance to Dox and the mutations caused by total loss-of-function of apoptotic signaling pathways (which required exclusion of the sample AF98b from the group with genetic alterations) gave a more significant correlation (*p*-value = 0.006).

The data obtained are important for further determination of the therapeutic strategy for STS patients with mutations in apoptosis signaling. Current approaches in the therapy of tumors with mutated *TP53* could be based either on the restoration of TP53 expression or the elimination of mutant TP53 [39,40]. Thus, a combination of APR-246 and azacytidine is involved in clinical trials of myelodysplastic syndromes with mutated *TP53* (NCT 03072043). APR-246 interacts with cysteine residues in mutant TP53. resulting in thermodynamical stabilization of the protein and normalization of function [41]. Additionaaly, Lee et al. demonstrated that tumor cell resistance to Dox, induced by TP53 mutations, is mediated by STAT3 activation, which could be decreased by Src inhibitors, in particular, saracatinib [13].

Although all chemotherapeutics used in the present study are genotoxic agents, their mechanisms differ significantly, which might explain the lack of association between the revealed genetic abnormalities in apoptotic signaling and resistance to Ifo, Doc, and Gem.

## 5. Conclusions

Taken together, our results indicate that the simultaneous performance of CSRA on primary STS cultures with the analysis of genetic alterations in the same cultured cells provides a novel approach for studying the MDR mechanisms in cancer cells. We have demonstrated that genetic abnormalities of apoptotic signaling pathways are observed in 27% analyzed STS and that these abnormalities are associated with chemoresistance to Dox.

## Figures and Tables

**Figure 1 cancers-14-01796-f001:**
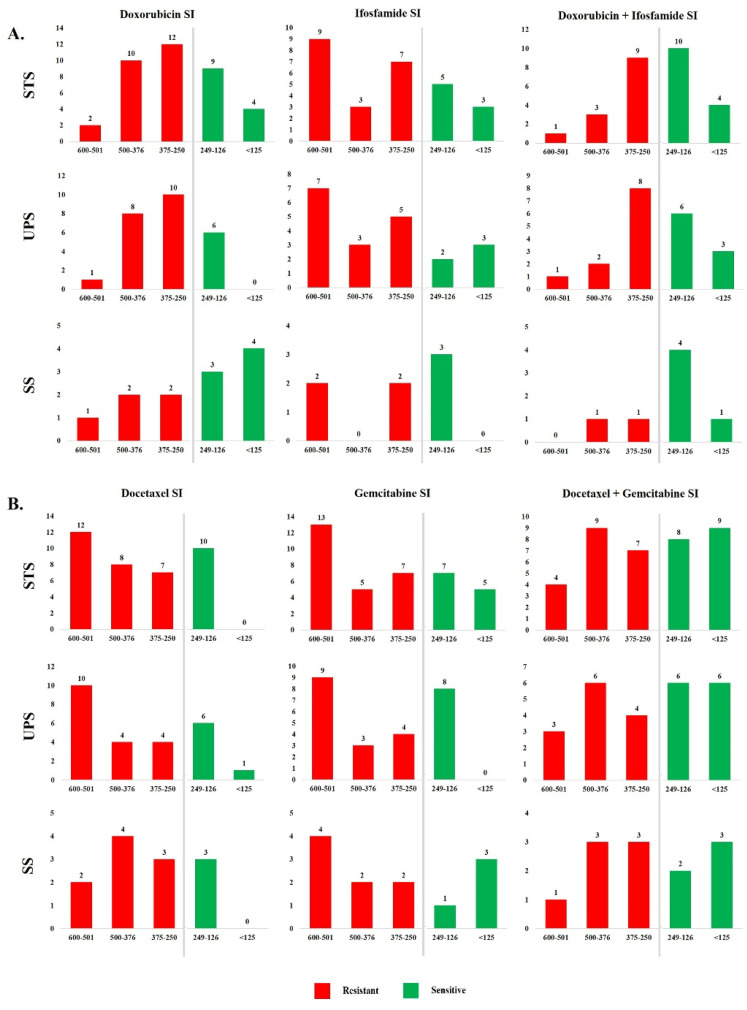
Histograms of SI frequencies of STS primary cultures for chemotherapeutic drugs and their combinations. (**A**). Number of SS, UPS, and STS primary cultures in the groups formed by SI to Dox, Ifo, and DOX + IFO; (**B**). Number of SS, UPS, and STS primary cultures in the groups formed by SI to Doc, Gem, and Doc + Gem. Red color indicates resistance; green color indicates sensitivity to the drugs.

**Figure 2 cancers-14-01796-f002:**
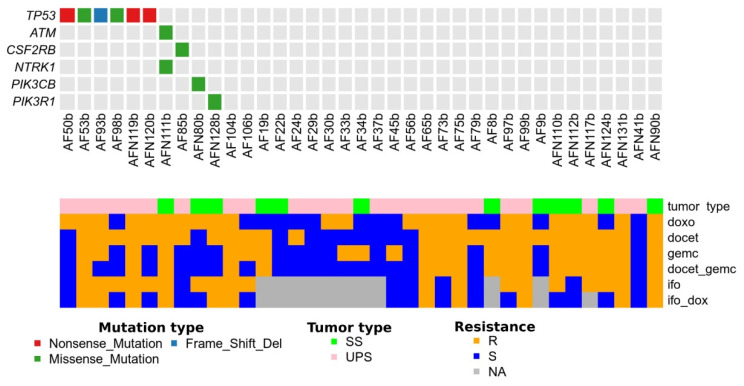
Data on resistance/sensitivity of STS specimens from a studied cohort of patients with/without structural alterations of genes associated with apoptosis induction, according to KEGG_APOPTOSIS. SS—synovial sarcoma, UPS—undifferentiated pleomorphic sarcoma, R—resistant STS, S—sensitive STS, NA—not available.

**Figure 3 cancers-14-01796-f003:**
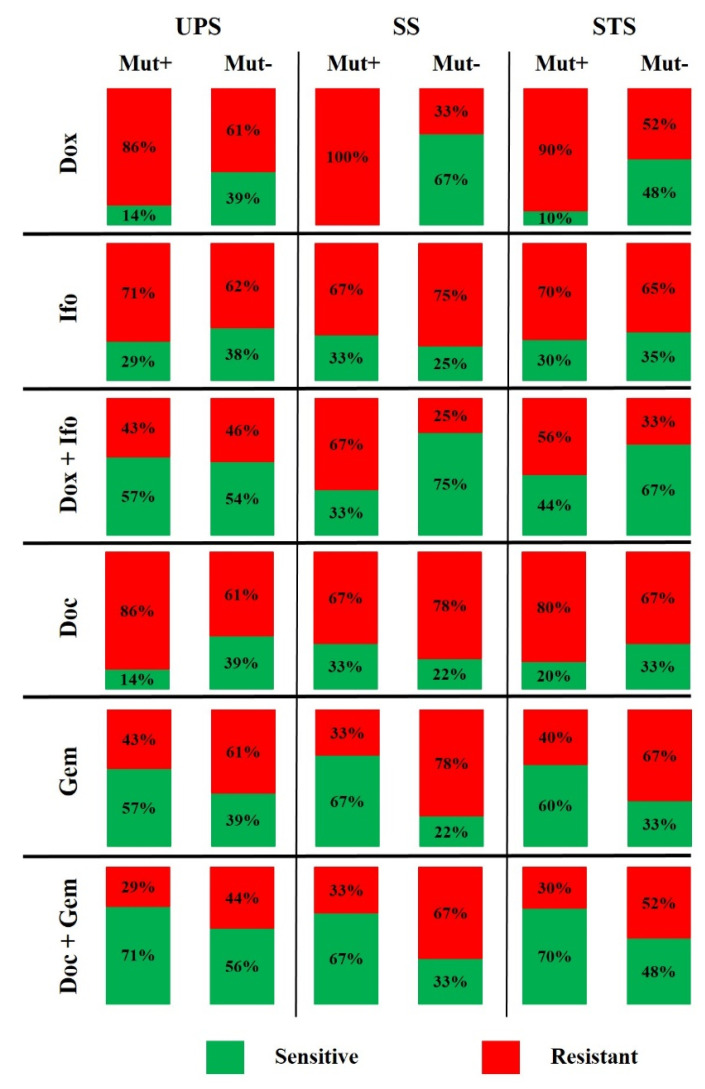
Histograms of partial ratio (%) of sensitivity and resistance to the drug primary STS cultures depending on the somatic mutational status of apoptosis activation pathway genes.

**Table 1 cancers-14-01796-t001:** Drugs tested and their 100% TDC as used in CSRA.

Drug/Combination	100% TDC (mg/mL)
Doxorubicin	1.0
Ifosfamide (4-hydroxy-ifosfamide)	3.0
Doxorubicin + Ifosfamide	1.0 + 3.0
Docetaxel	11.3
Gemcitabine	25.0
Docetaxel + Gemcitabine	11.3 + 25.0

**Table 2 cancers-14-01796-t002:** Clinical characteristics of patients.

Characteristics	Percentage, %
Histological subtype SS UPS	12 (32%) 25 (68%)
Age <40 >40	12 (32%) 25 (68%)
Sex Male Female	19 (51%) 18 (49%)
Tumor grade G1–G2 G3	2 (6%) 35 (94%)
Stage I–II III–IV	8 (22%) 29 (78%)
Newly diagnosed Recurrent	23 (62%) 14 (38%)
NeoCT NeoRT Without neoadjuvant therapy	20 (54%) 3 (8%) 14 (38%)

**Table 3 cancers-14-01796-t003:** Structural alterations in genes from KEGG_APOPTOSIS signaling revealed in STS specimens of studied cohort of samples.

No of Specimen	Gene	Specimen Code in Sequencing	Gene Mutation	Protein Mutation	Cosmic Database ID
AF50b	*TP53*	tumor_50.GRCh38DH.exome	g.chr17:7675234G>C	p.Y126 *	COSM10862
AF53b	*TP53*	tumor_53.GRCh38DH.exome	g.chr17:7674954G>A	p.H193Y	COSM10672
AF93b	*TP53*	tumor_93.GRCh38DH.exome	g.chr17:7674191delC	p.E258fs	COSM7340859
AF98b	*TP53*	tumor_98.GRCh38DH.exome	g.chr17:7670678A>C	p.L344R	COSM46303
AFN119b	*TP53*	tumor_N119.GRCh38DH.exome	g.chr17:7670685G>A	p.R342 *	COSM11073
AFN120b	*TP53*	tumor_N120.GRCh38DH.exome	g.chr17:7675218T>A	p.K132 *	COSM44641
AF85b	*CSF2RB*	tumor_85.GRCh38DH.exome	g.chr22:36936630A>G	p.S516G	
AFN111b	*ATM*	tumor_N111.GRCh38DH.exome	g.chr11:108335029C>T	p.R2691C	COSM922745
	*NTRK1*		g.chr1:156879231G>T	p.V639L	
AFN128b	*PIK3R1*	tumor_N128.GRCh38DH.exome	g.chr5:68294573A>G	p.N125S	Analogue of COSM6960758
AFN80b	*PIK3CB*	tumor_N80.GRCh38DH.exome	g.chr3:138714521T>C	p.T417A	Analogue of COSM419799

* nonsense mutation.

**Table 4 cancers-14-01796-t004:** Chemoresistance of cancer cells in the primary STS cultures depending on the somatic mutational status of apoptosis activation pathway genes *.

Drug	Characteristics	UPS		SS		STS	
		25		12		37	
		Mut+	Mut−	Mut+	Mut−	Mut+	Mut−
		7	18	3	9	10	27
**Dox**	**Sensitive (n)**	1	7	0	6	1	13
	**Resistant (n)**	6	11	3	3	9	14
	***p*-value**	0.25		0.38		**0.036 ***	
**Ifo**	**Sensitive (n)**	2	5	1	1	3	6
	**Resistant (n)**	5	8	2	3	7	11
	***p*-value**	0.52		0.71		0.56	
**Dox + Ifo**	**Sensitive (n)**	4	7	1	3	4	12
	**Resistant (n)**	3	6	2	1	5	6
	***p*-value**	0.63		0.37		0.24	
**Doc**	**Sensitive (n)**	1	7	1	2	2	9
	**Resistant (n)**	6	11	2	7	8	18
	***p*-value**	0.25		0.62		0.36	
**Gem**	**Sensitive (n)**	4	7	2	2	6	9
	**Resistant (n)**	3	11	1	7	4	18
	***p*-value**	0.35		0.24		0.14	
**Doc + Gem**	**Sensitive (n)**	5	10	2	3	7	13
	**Resistant (n)**	2	8	1	6	3	14
	***p*-value**	0.40		0.36		0.21	

* “Mut+”—at least one of the genes of apoptosis signaling is mutated, “Mut−”—mutations in genes of apoptosis signaling were not found, *p*-value in bold shows significant association of the drug resistance and genetic abnormalities in apoptosis signaling.

## Data Availability

The data presented in this study are available on request from the corresponding author.

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
