# Peer review of "Soft Tissue Sarcoma Study: Association of Genetic Alterations in the Apoptosis Pathways with Chemoresistance to Doxorubicin"

_cancers, 2022, doi:10.3390/cancers14071796_

Round 1

Reviewer 1 Report

In this paper, Kirilin et al describe their study on chemotherapy resistance in soft tissue sarcomas. Authors used an in vitro model of patient derived cell lines and treated these with multiple types of chemotherapy, followed by mutation analysis of cell lines. They found that alterations in the TP53 pathway were associated with resistance to Doxorubicin chemotherapy. 

Major concerns:

  • The introduction needs improvement to increase its readability and flow. I would suggest to start the paragraphs with an introductory sentence and to elaborate a bit more on TP53-mediated apoptosis, so that the results that involve this pathway are easier to understand.
  • Methods are well described, but some major details are lacking. Readers might be interested in details of controls that were used in the chemosensitivity assay. And: were the primary cell lines confirmed to be sarcoma cells, and how was this done?
  • The discussion is quite short; I would recommend a more extensie discussion of the relevance of the data, strengths and weaknesses of their study and suggestions for further research. In the last part of the result section (line 196 - line 231), most of the results of the genetic analysis are discussed already – I would suggest to move the explanation of these results to the discussion section.
  • It would really strengthen the paper if the results could be validated in a clinical cohort

Minor concerns:

Lines 21-22, 36: Is it drug resistance that develops during treatment or do some tumors not respond at all?

Line 28 and further: ‘Gemcytabin’ should be ‘Gemcitabine’

Line 52: remove ‘wide’ or ‘heterogeneous’; since their meaning is the same in this context

Line 54: Please provide time period of this incidence and population (worldwide?)

Line 56: Chemotherapy is not obligatory in STS treatment. The mainstay of (curative) treatment is surgery, chemotherapy is sometimes used for high grade sarcomas in (neo-)adjuvant setting and but mostly stage 4 disease.

Line 58: ‘has reached only 40-60%...’ - missing reference

Line 78: ‘Leomyosarcoma’ should be ‘Leiomyosarcoma’

Line 83: ‘TP53-mediated … of apoptotic signaling.’ - missing reference or explanation

Lines 91-97: I would suggest to move the sentence ‘we used …. for estimation of cell viability’ to the methods section

Line 92: Is there a rationale to choose for those subtypes?

Line 101-102: I would suggest to change this to: ‘between 2018-2020’. Did those patients undergo treatment (chemotherapy, radiotherapy) before surgery? This is important to know for the interpretation of the results on chemo sensitivity.

Line 150: It seems that the success rate of establishing cell lines from tumor tissue is 100%, but that seems unrealistic.  

Line 122: How was the SI calculated for the drug combinations? Were TDCs of the combination treatment similar to concentrations used for monotherapy? How were these concentrations combined?

Line 165: Please enlarge graphs to make it more clear. Synovial sarcoma and undifferentiated sarcoma are different entities that could have variable response to chemotherapy; so it would be better to show the results of this assay for both subtypes separately. 

Line 170: It is unclear what is meant by ‘KEGG_APOPTOSE’

Line 188-189: Meaning of ‘Association of tumor cell resistance to Dox was not demonstrated for UPS and SS.’ is unclear.

Lines 189-190: ‘TP53 mutations are more specific to UPS (6 out of 7)’  is not correct according to figure 2 and table 2 - all TP53 mutations that were found were in 6 UPS samples.

Table 3: The meaning of the p-values is unclear, I would suggest to add an explanation of the p-values and to explain the definition of ‘Mut +’ ‘Mut –‘  does Mut + mean that at least one of the genes is mutated?

Author Response

Dear Peer Reviewer,

We are very grateful for your careful reading our manuscript, its thorough analysis and useful comments. We appreciate your rational pieces of advice very much and did our best to improve our manuscript following your recommendations. Please find our responses on all your comments point by point.

Major comment 1: The introduction needs improvement to increase its readability and flow. I would suggest to start the paragraphs with an introductory sentence and to elaborate a bit more on TP53-mediated apoptosis, so that the results that involve this pathway are easier to understand.

Response 1: We have changed Introduction by paying more attention both to P53-mediated apoptosis and apoptosis signaling on the whole.

Major comment 2: Methods are well described, but some major details are lacking. Readers might be interested in details of controls that were used in the chemosensitivity assay. And: were the primary cell lines confirmed to be sarcoma cells, and how was this done?

Response 2: We have changed the corresponding parts of the Material and method section describing cytology analysis of primary cultures.

Major comment 3: The discussion is quite short; I would recommend a more extensive discussion of the relevance of the data, strengths and weaknesses of their study and suggestions for further research. In the last part of the result section (line 196 - line 231), most of the results of the genetic analysis are discussed already – I would suggest to move the explanation of these results to the discussion section.

Response 3: We have followed your advice to move the explanation of the results to the discussion section.

Major comment 4: It would really strengthen the paper if the results could be validated in a clinical cohort

Response 4: We focused our present study on the analysis of the genetic alteration role in the development of chemoresistance by comparing results of in vitro CSRAs and mutations in apoptosis signaling and the period after the treatment is still rather short for some patients. A year or two years later it will be possible to extend this study with clinical data of the patient status.

Minor comment 1: Lines 21-22, 36: Is it drug resistance that develops during treatment or do some tumors not respond at all?

Response 1: Our results that some STS tumors do not respond at all, when they are treated in culture with genotoxic agents, are in accordance with the common point of view that STS tumors are often resistant to genotoxic anticancer therapy. In the text of our work we provide the corresponding reference on the paper of Judson et al., (doi: 10.1016/S1470-2045(14)70063-4). Moreover, primary cell culture cultivation time with cytostatic was too short to enable the selection of resistant cells and their multiplication.

Minor comment 2: Line 28 and further: ‘Gemcytabin’ should be ‘Gemcitabine’

Response 2: We have corrected the mistake.

Minor comment 3: Line 52: remove ‘wide’ or ‘heterogeneous’; since their meaning is the same in this context.

Response 3: We have corrected the text.

Minor comment 4: Line 54: Please provide time period of this incidence and population (worldwide?)
Response 4: We corrected the text.

Minor comment 5: Line 56: Chemotherapy is not obligatory in STS treatment. The mainstay of (curative) treatment is surgery, chemotherapy is sometimes used for high grade sarcomas in (neo-)adjuvant setting and but mostly stage 4 diseases.

Response 5: We have corrected the text.

Minor comment 6: Line 58: ‘has reached only 40-60%...’ - missing reference

Response 6: We have detailed the text and added the reference.

Minor comment 7: Line 78: ‘Leomyosarcoma’ should be ‘Leiomyosarcoma’

Response 7: We have corrected the mistake.

Minor comment 8: Line 83: ‘TP53-mediated … of apoptotic signaling.’ - missing reference or explanation

Response 8: We have added the references.

Minor comment 9: Lines 91-97: I would suggest to move the sentence ‘we used …. for estimation of cell viability’ to the methods section.

Response 9: The last paragraph of the introduction is devoted to our approaches, not to the methodology description. By this paragraph we would like to emphasize that analogous studies directed to the analysis of chemoresistance may be performed using different protocols, but following the approach, elaborated by Kurbacher et al. That is why we would like to request you not to move this paragraph to the methodology section.

Minor comment 10: Line 92: Is there a rationale to choose for those subtypes?

Response 10: There is no information concerning genome alterations associated with apoptosis signaling for these types of STS. It was the reason why we considered these STS subtypes.

Minor comment 11: Line 101-102: I would suggest to change this to: ‘between 2018-2020’. Did those patients undergo treatment (chemotherapy, radiotherapy) before surgery? This is important to know for the interpretation of the results on chemo sensitivity.

Response 11: For a number of patients, the development of resistance of tumor cells during CSRA was observed after neoadjuvant therapy, but this is not the only condition for resistance appearance. The sample size does not allow us to assess how important this factor was.

Minor comment 12: Line 150: It seems that the success rate of establishing cell lines from tumor tissue is 100%, but that seems unrealistic.

Response 12: Our study was limited by exome sequencing as it is rather expensive analysis. As concerns rate of establishing cell lines from tumor tissue, we got 211 primary STS cultures (cytology analysis was performed for all of them), but 259 tumor samples were taken for it. So, the efficiency of establishing cell lines from tumor tissue is 81%. We have added this information in the methodology section.

Minor comment 13: Line 122: How was the SI calculated for the drug combinations? Were TDCs of the combination treatment similar to concentrations used for monotherapy? How were these concentrations combined?
Response 13: In the combination treatment we used concentrations similar to ones used for monotherapy as the analogous regimes are recommended for combined chemotherapy of STS patients respectively to monotherapy.

Minor comment 14: Line 165: Please enlarge graphs to make it more clear. Synovial sarcoma and undifferentiated sarcoma are different entities that could have variable response to chemotherapy; so it would be better to show the results of this assay for both subtypes separately. 
Response 14: We have detailed the histogram.

Minor comment 15: Line 170: It is unclear what is meant by ‘KEGG_APOPTOSE’
Response 15: We have provided URL for this database of apoptosis signaling.

Minor comment 16: Line 188-189: Meaning of ‘Association of tumor cell resistance to Dox was not demonstrated for UPS and SS.’ is unclear.

Response 16: We have corrected the phrase: “Association of tumor cell resistance to Dox with genetic alterations in apoptosis signaling was not demonstrated for UPS and SS”.

Minor comment 17: Lines 189-190: ‘TP53 mutations are more specific to UPS (6 out of 7)’ is not correct according to figure 2 and table 2 - all TP53 mutations that were found were in 6 UPS samples.

Response 17: We have observed 7 mutations in UPS: 6 mutations in TP53 and 1 mutation in CSF2RB. It means that TP53 mutations are more frequent in UPS. We have corrected the text.

Minor comment 18: Table 3: The meaning of the p-values is unclear, I would suggest to add an explanation of the p-values and to explain the definition of ‘Mut +’ ‘Mut –‘ à does Mut + mean that at least one of the genes is mutated?

Response 18: We have described Mut +’ ‘Mut –‘ and p-value: “* “Mut +” – at least one of the genes of apoptosis signaling is mutated, “Ðœut -“  – mutations in genes of apoptosis signaling were not found, p-valuse show significance of the association of drug resistance and genetic abnormalities in apoptosis signaling”, and added Figure 4 with the histogram of the Histograms of partial ratio (%) of sensitive and resistant to the drug primary STS cultures in dependence on the somatic mutational status of apoptosis activation pathway genes.

Reviewer 2 Report

The submitted manuscript by Evgeny M et al., titled “Soft Tissue Sarcoma Study: Association of Genetic Alterations 2 in the TP53-mediated Pathway for Apoptosis Activation with 3 Chemoresistance to Doxorubicinis an interesting article. In this study, authors have demonstrated that genetic alterations in TP53, ATM, PIK3CB, PIK3R1, NTRK1, and CSF2RB genes involved in apoptosis pathways are associated with doxorubicin chemoresistance in STS primary culture. It would be helpful for the readers if authors can address the following concerns.

  1. Why do authors just focus on the genetic alterations of TP53-mediated apoptotic pathway genes? The authors should explain this point in the introduction and discussion. What are the other alterations observed by sequencing?  
  2. In figure. 2, multiple STS primary cells with wild-type p53 are resistant to Doxorubicin, what is the explanation for dox resistance in these cells? It seems other genetic alterations are driving resistance like RTKs, etc.
  3. Authors should do knockdown/knockout of wild-type p53 in primary cells and then treat with the drugs to confirm the role of p53 in dox resistance.

Author Response

Dear Peer Reviewer,

We are very grateful for your reading our manuscript, its analysis and useful comments. Please find our responses on your comments.

Comment 1: Why do authors just focus on the genetic alterations of TP53-mediated apoptotic pathway genes? The authors should explain this point in the introduction and discussion. What are the other alterations observed by sequencing?  

Response 1: The main goal of our study was the analysis of the association of STS chemoresistance to the drugs recommended for first- and second-line therapy with genetic abnormalities, so that to personalize STS therapy and to reduce STS patient exposure to the useless toxic treatment. In different cancer types the drugs used in our study promote the development of MDR through the activation of ABC-transporters, however Oda et al. (doi: 10.1002/ijc.20589) demonstrated that the expression of ABC-transporters in STS is a very rare event and probably does not have clinical significance. Our recent results, which have been recently accepted for publication by IJMS, concerning ABC-transporter expression in STS are in agreement with the data of Oda et al. As concerns TP53, this gene was shown by Lazar et al. to represent a significantly mutated gene in STS (doi: 10.1016/j.cell.2017.10.014) and the mechanism of TP53 mutation realizing was studied by Lee et al. (doi:10.1177/1010428318794217), who demonstrated that TP53 mutations cause activation of STAT3 and, by this way, inactivate apoptosis. As inactivation of apoptosis signaling may be caused not only by TP53 mutations, but also by mutations in the other components of apoptosis signaling we focused our study on the genetic abnormalities of all the components of apoptosis signaling, presented at the database KEGG_APOPTOSIS (https://www.genome.jp/pathway/hsa04210).

Comment 2: In figure. 2, multiple STS primary cells with wild-type p53 are resistant to Doxorubicin, what is the explanation for dox resistance in these cells? It seems other genetic alterations are driving resistance like RTKs, etc.

Response 2: Our study was focused on the search of genetic abnormalities in apoptosis signaling in STS and analysis of their associations with chemoresistance to the drugs recommended for STS first- and second-line therapy to translate this knowledge to clinical practice to choose more optimal chemotherapeutic courses in dependence on genetic analysis without patient exposure to the useless toxic treatment. Takin into consideration your comment, we added in the discussion the description of one of the possible mechanisms of the influence of TP53 mutations on apoptosis, which has been already studied by Lee et al. (doi:10.1177/1010428318794217). They demonstrated that TP53 mutations cause activation of STAT3 and, by this way, inactivate apoptosis. As concerns the cells resistant to chemotherapy but not harboring mutations in the genes of apoptosis signaling, according our data in press, one of the possible mechanism of chemoresistance may be represented by activated expression of MVP (This gene encodes the major component of the vault complex, which is involved in nucleo-cytoplasmic transport).

Comment 3: Authors should do knockdown/knockout of wild-type p53 in primary cells and then treat with the drugs to confirm the role of p53 in dox resistance.

Response 3: Role of p53 mutations in apoptosis inactivation has been already demonstrated by Lee et al. (doi:10.1177/1010428318794217), We added this information in the discussion.

Reviewer 3 Report

Kirilin and colleagues present a refreshing, honest and accurate assessment of the role of TP53 mutations in chemotherapy sensitivity in primary syno-vial sarcoma (SS) and undifferentiated pleomorphic sarcoma (UPS). Using fresh surgical samples, the authors use cells from these tumors for chemo-sensitivity assays and then correlate these results with genome sequencing of matched tumors. The data presented in this manuscript is well controlled and provide real clinical clarity. I have only minor comments that are designed to improve the readability and impact of this research.

  1. I would like to see a graph of MUT vs. WT Dox sensitivities values for UPS, SS and STS in table 3. I think this will more clearly illustrate the data.
  2. I think that resistance and sensitive should be clearly delineated on the graphs presented in figure 1.
  3. Although, the title is accurate, it's pretty long. I would shorten it.
  4. The discussion is pretty short and focused. I enjoyed reading it, but I feel that it requires expansion and better integration with the field.

Author Response

Dear Peer Reviewer,

We are very grateful for your reading our manuscript and its thorough analysis. We appreciate your rational pieces of advice very much and did our best to improve our manuscript following your recommendations. Please find our responses on all your comments.

Comment 1: I would like to see a graph of MUT vs. WT Dox sensitivities values for UPS, SS and STS in table 3. I think this will more clearly illustrate the data.

Response 1: We liked very much your suggestion, thank you! We have added the histograms of partial ratio (%) of sensitive and resistant to the drug primary STS cultures in dependence on the somatic mutational status of the genes of apoptosis activation pathways.

Comment 2: I think that resistance and sensitive should be clearly delineated on the graphs presented in figure 1.

Response 2: We have done it.

Comment 3: Although, the title is accurate, it's pretty long. I would shorten it.

Response 3: We have changed the title according to the comments of the 1st reviewer and it becomes a little bit shorter.

Comment 4: The discussion is pretty short and focused. I enjoyed reading it, but I feel that it requires expansion and better integration with the field.

Response 4: The discussion has been extended.

Round 2

Reviewer 1 Report

After reading the revised version, my advise is to accept.

Reviewer 2 Report

The current version of the manuscript is improved and suitable for publication.